# ConfounderGAN: Protecting Image Data Privacy with Causal Confounder

**Qi Tian** [1]    **Kun Kuang** [1,5,*]  **Kelu Jiang** [1]    **Furui Liu** [2]    **Zhihua Wang** [3]    **Fei Wu** [1,3,4,5]

[1] College of Computer Science and Technology, Zhejiang University, Hangzhou, China
[2] Huawei Noah's Ark Lab, Beijing, China
[3] Shanghai Institute for Advanced Study of Zhejiang University, Shanghai, China
[4] Shanghai AI Laboratory, Shanghai, China
[5] Key Laboratory for Corneal Diseases Research of Zhejiang Province, Hangzhou, China
{tianqics,kunkuang,jiangkelu,zhihua.wang,wufei}@zju.edu.cn

## Abstract

The success of deep learning is partly attributed to the availability of massive data downloaded freely from the Internet. However, it also means that users' private data may be collected by commercial organizations without consent and used to train their models. Therefore, it's important and necessary to develop a method or tool to prevent unauthorized data exploitation. In this paper, we propose *ConfounderGAN*, a generative adversarial network (GAN) that can make personal image data unlearnable to protect the data privacy of its owners. Specifically, the noise produced by the generator for each image has the confounder property. It can build spurious correlations between images and labels, so that the model cannot learn the correct mapping from images to labels in this noise-added dataset. Meanwhile, the discriminator is used to ensure that the generated noise is small and imperceptible, thereby remaining the normal utility of the encrypted image for humans. The experiments are conducted in six image classification datasets, consisting of three natural object datasets and three medical datasets. The results demonstrate that our method not only outperforms state-of-the-art methods in standard settings, but can also be applied to fast encryption scenarios. Moreover, we show a series of transferability and stability experiments to further illustrate the effectiveness and superiority of our method.

## 1 Introduction

In recent years, deep learning has achieved great success in many fields, such as computer vision [10], natural language processing [6], *etc*. An important factor is that large-scale datasets provide rich and diverse training examples for deep neural networks. However, many datasets are collected from some free channels without mutual consent, and this unauthorized collection of private data may violate the rights of the data owner [3]. For example, Kashmir Hill from the New York Times recently reported that a company, named Clearview.AI, used web crawler scripts to collect more than 3 billion online photos from various multimedia websites (*e,g.*, Facebook) for training its own commercial models [35]. This large unauthorized dataset goes far beyond anything ever constructed by the United States government or Silicon Valley giants. In addition, Google's Nightingale Project has also been reported to collect medical care information from tens of millions of patients without their knowledge, and use advanced machine learning technology to customize medical services for individual patients [23, 37].

In this context, various information privacy laws around the world have been published to describe the rights of natural persons to control who is using its data [41]. *e.g.*, General Data Protection

---

*Corresponding author.

Regulation (GDPR) in the European Union [39], Electronic Communications Privacy Act (ECPA) in the U.S. [40, 38], *etc*. However, in addition to legal constraints, it's necessary to develop a protection method to actively prevent unauthorized data from being used by third parties to train commercial models. In this paper, we aim to develop an image encryption method. *i.e.*, the deep classification models cannot extract any exploitable knowledge from encrypted (or processed) images. Moreover, we hope the modification of the original data is strictly limited, thereby retaining the data quality for normal usage. *e.g.*, an unlearnable photo should be free from obvious visual defects so it can be shared with friends on social media.

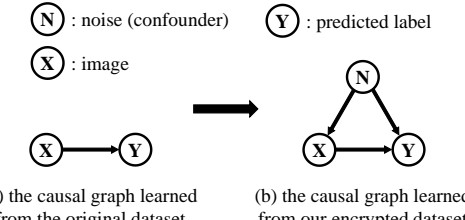

Figure 1: Causal graphs learned from the original dataset and our encrypted dataset.

Unfortunately, the existing works are very sparse and have application limitations. Shan et al. [30] first propose an error-maximizing noise based on metric learning to make data unexploitable, but this method is mainly customized for face recognition systems. Then Huang et al. [12] propose a bi-level optimization based on error-minimizing noise (EMN) to meet the above goal, and it can be extended to various image datasets. However, these methods need multiple gradient backpropagations to generate an effective noise for each labeled image, which means they cannot encrypt an unlabeled image on the fly. We emphasize that this setting is common and important for real-world applications. For example, a person takes a photo with a cell phone. Then he/she wants to quickly encrypt the photo and share it on social media immediately.

In this paper, we propose a novel image data encryption method called *ConfounderGAN* to solve the above challenges. Figure 1 demonstrates the core idea of our method. We suppose that the model can learn a correct causal graph from the image $X$ to the label $Y$ in the original dataset as illustrated in Figure 1(a). To prevent unauthorized models from extracting ground truth relationship $X \to Y$ from the private dataset, we deliberately introduce a confounder noise $N$ that satisfies the relationship $X \leftarrow N \to Y$ in the dataset as shown in Figure 1(b). This confounder noise is correlated with both the image $X$ and the label $Y$, creating a spurious correlation between $X$ and $Y$. Therefore, models trained on this data would fit the negatively spurious correlation and get low test accuracy. There are many possible ways that can achieve this confounder-based encryption framework. We instantiate this framework by generative adversarial networks (GAN) [9], as this implementation is more efficient and practical. Specifically, we hope that the generator can produce confounder noise that satisfies the above causal graph for each private image. The main challenge is how to ensure that the noise is strongly associated with both the image and the label. Since the noise is produced by the generator with the original image as input, the former is naturally satisfied. For the latter, we directly classify the noise as the ground truth class of the original image during generator training. Meanwhile, we send the original image and its noise-added copy to the discriminator, thus ensuring the invisibility of noise. Once the generator is trained, it can quickly produce confounder noise for any image to protect its privacy, including those unlabeled ones. Our main contributions are:

- We present a confounder-based framework for image data encryption. The confounder in this framework can create spurious correlations between images and labels, thus preventing unauthorized models from learning exploitable knowledge on the encrypted data.

- By leveraging the property of generative adversarial networks, we implement the above framework as ConfounderGAN, where the generator is used to produce confounder noise to encrypt the target image and the discriminator ensures that the noise is imperceptible.

- Extensive experiments on six image classification datasets show that our proposed method not only outperforms state-of-the-art methods in standard settings, but can also be applied to fast encryption scenarios. Also, we conduct a set of transferability and stability experiments to highlight the effectiveness and superiority of our method.

## 2 Related work

In this section, we briefly review some literature that is closely related to our work: data privacy, data poisoning, causal confounder in deep learning.

**Data privacy.** Privacy-preserving techniques have been widely explored in the machine learning community and a lot of excellent works have been proposed to protect data privacy [32, 26, 27, 33]. However, these methods are mainly to prevent malicious agencies from de-inferring training set information in a trained model. In this paper, we focus on a more challenging scenario. *i.e.*, making private data completely unlearnable by unauthorized deep neural networks. Shan et al. [30] first propose a solution for the face recognition system. By utilizing the targeted adversarial attack to generate error-maximizing noise, this method can close the representation of different identities so that the model trained on this dataset can only obtain poor performance. Then Huang et al. [12] propose a bi-level optimization based on error-minimizing noise (EMN) to meet the same goal, and it is a general method that can be used to various image datasets. However, these gradient-based methods need multiple optimization steps for encrypting one labeled image, and thus cannot quickly encrypt an unlabeled one. Instead, our ConfounderGAN does not have this limitation. Because once the confounder noise generator is trained, any private image can be processed immediately.

**Data poisoning.** The goal of traditional data poisoning is to reduce the test accuracy of the model by modifying the training set. Biggio et al. [2] first introduce this type of attack in support vector machines. Then Muñoz-González et al. [21] propose a deep learning version by poisoning the most representative samples in the training examples. Although data poisoning attacks seem to share the same goal as ours, these methods have a limited impact on DNNs and are not suitable for data protection tasks. *e.g.*, the model trained on poisoned examples can still achieve acceptable performance [21], and the modified image can be easily distinguished from the original one [44]. In addition, the backdoor attack is another type of attack that poisons training data with a trigger pattern [4, 29, 20], but this attack does not prevent the model from learning useful knowledge in the natural data. Therefore, traditional data poisoning methods cannot be used for data protection, while our proposed method can produce unlearnable examples with imperceptible noise.

**Confounder in deep learning.** In statistics, a confounder is a variable that influences both the cause variable and effect variable, causing a spurious association [24]. If there are confounders in the training set, the model may be hard to learn the correct causal relationship from input to output, resulting in sub-optimal performance [17, 36]. A lot of works have been proposed to mitigate the negative impact of confounders in deep learning [31, 42, 43, 18]. *e.g.*, Shen et al. [31] believe that the background in the dataset plays the role of a confounder and they propose to improve by sample reweighting. However, all these methods treat the confounder as a defect. Instead, our proposed method utilizes the property of the confounder to achieve data privacy protection.

## 3 Problem statement

In this paper, we focus on the scenario of image data encryption. There are two characters in this scenario: data owner and unauthorized model trainer. The data owner needs to encrypt their private data to prevent the model trainer from obtaining the high-performance classification model on this data. We consider two practical encryption settings:

(1) In-distribution data encryption (standard setting)

This setting follows [30, 12]. The data owner has $n$ labeled natural (unencrypted) data $\mathcal{D}_{nat}$. He/she can actively use a protection method to obtain encryption (unlearnable) images by adding imperceptible perturbation to the original images. Since the protection method has access to each data to adaptively learn and generate custom unlearnable noises, these processed data are defined as in-distribution encrypted data $\mathcal{D}_{in,en}$.

(2) Out-of-distribution data encryption (fast encryption setting)

The data owner trains a cryptographic tool (*e.g.*, a generator) with labeled historical data. Once the encryption tool is trained, it can be used for fast encryption on any data. Specifically, if the data owner uses this tool to encrypt the historical data (*i.e.*, the training set of the encryption tool), the processed data is denoted as $\mathcal{D}_{in,en}$. Instead, if they use the tool to encrypt some newly unlabeled data (*i.e.*, the non-training set of the encryption tool), the processed data is called out-of-distribution encrypted data $\mathcal{D}_{out,en}$. We believe this setting is practical for the real world. *e.g.*, a person can install the encryption app on the phone in advance, and then their instant photos can be quickly encrypted by this app.

One unauthorized model trainer can download any data from public sources as the training set. *i.e.*, $\mathcal{D}_{tr} = \{\mathcal{D}_{in,en}, \mathcal{D}_{out,en}, \mathcal{D}_{nat}\}$, where $\mathcal{D}_{nat}$ is collected means that not everyone would encrypt

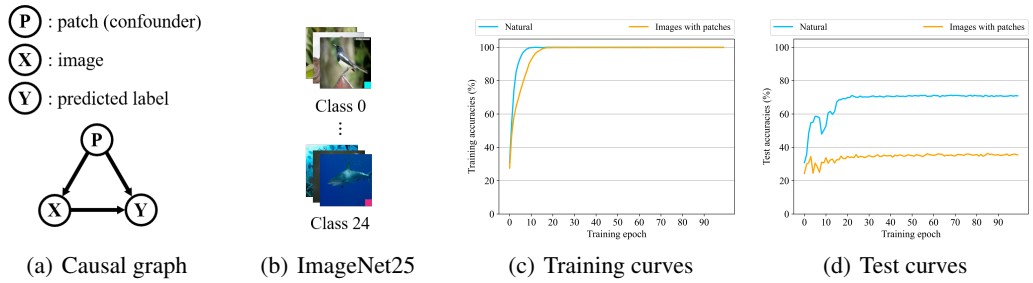

Figure 2: Introducing confounders in ImageNet25 by class-wise patches.

their data before publishing. Then the trainer can label the collected examples $x$ with classes $t$ and train the model $h$ with cross-entropy loss $\ell$, as

$$\mathbb{E}_{x \sim \mathcal{P}_{data}} \ell(h(x), t),\tag{1}$$

where $\mathcal{P}_{data}$ represents the distribution of the training set. Note that data encryptors cannot predict the information of model training in advance (*e.g.*, model backbone, data augmentation), which poses a challenge to the encryption method.

Finally, we measure the effectiveness of the unlearnable example by evaluating the accuracy of model $h$ on the standard test set $\mathcal{D}_{test}$. Intuitively, low test accuracy indicates that the model cannot learn exploitable knowledge from the encrypted dataset.

## 4 Methodology

In this section, we first demonstrate how the native confounder (*i.e.*, background) in the image data affects the performance of a classification model. Inspired by this phenomenon, we introduce a general framework for image data encryption with the causal confounder. Then we instantiate this framework as ConfounderGAN.

### 4.1 Motivation

We observe that the background in some datasets may play the role of the confounder in model learning, thus reducing model performance. Specifically, the image classification task can be modeled as a causal graph. The input image is the cause $X$, and the output prediction is the effect $Y$. The training process can be regarded as the process of using the ground truth information to regress relationships from images to predictions $X \rightarrow Y$. Ideally, if the model can properly learn these true causal relationships, the classification accuracy will be optimal. However, current data-driven deep learning techniques are only good at learning correlations rather than causality, thus the co-occurrence frequency of object and background may influence the model learning *e.g.*, if the dog is always on the grass in the training set, the model may use grass as a key feature of the class dog instead of the dog itself. In other words, the confounder background $B$ (*i.e.*, grass) constructs a spurious correlation between images and predictions $X \leftarrow B \rightarrow Y$ so that a biased model is learned. Nevertheless, we also need to be aware that native confounders (*e.g.*, background) in natural datasets do not play a dominant role in training, as models trained on this dataset can still achieve acceptable performance.

### 4.2 Confounder-based encryption framework

Inspired by the above observations, if we can deliberately introduce a confounder $C$ that is closely related to image $X$ and label $Y$ (*i.e.*, $X \leftarrow C \rightarrow Y$), the trained model would hardly learn the correct relationship $X \rightarrow Y$ from this dataset. We believe this is a general data encryption framework and there are many possible ways to implement it. For example, one straightforward approach is to add class-wise patches (CWP) to the training set. That is, the images of one class add the same patch, while the patches corresponding to each class are different. As shown in Figure 2(a), since the patch is added to the image, $P \rightarrow X$ is constructed. Meanwhile, the patch of each image is related to the image's class, so this builds the path $P \rightarrow Y$. To verify this idea, we select the first 25 classes

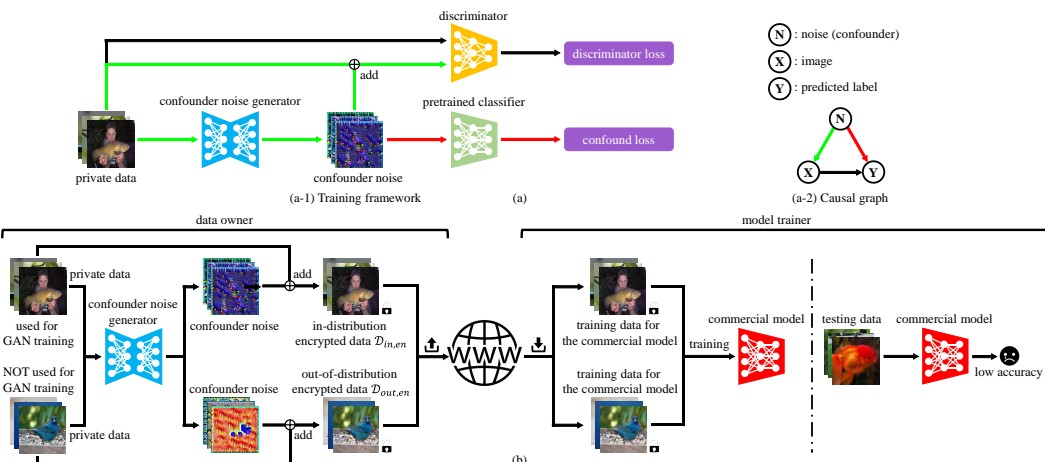

Figure 3: Pipeline of ConfounderGAN: (a) Training pipeline. (b) Evaluation pipeline.

from ImageNet to establish a tiny dataset ImageNet25 for experiments. As illustrated in Figure 2(b), each $224 \times 224$ image in the training set adds a $36 \times 36$ patch by class in the lower right corner. More experimental details are elaborated in Appendix G.2. Figure 2(c) and Figure 2(d) shows the training and test accuracy curves during model training, where the final test accuracies on the natural ImageNet25 and the modified ImageNet25 are 70.9% and 35.4%, respectively (Figure 2(d)). This demonstrates that such a simple confounder-based method suppresses the learnable knowledge in the dataset to some extent.

Unfortunately, the patch is too conspicuous in the image, affecting its natural use. A natural idea is to replace class-wise patches with imperceptible class-wise noises (CWN). However, this alternative can be circumvented by early stopping [12], *i.e.*, the model trainer can train a high-accuracy model on this noise-added dataset by early stopping. Meanwhile, both CWP and CWN have two important drawbacks: (1) The patch or the noise is predefined and not correlated with the image, so the relationship between the patch/noise (confounder) and image $X$ is not strong enough. (2) These methods are limited to in-distribution data encryption and cannot be applied to out-of-distribution data encryption. Hence, how to reasonably introduce a confounder into the training data to overcome the above drawbacks needs to be further explored.

### 4.3 ConfounderGAN:Confounder generation method based on GAN

In order to develop an effective encryption algorithm that can cover various practical scenarios, we utilize adversarial generative networks (GAN) [9] to instantiate the above framework as ConfounderGAN. Our goal is to train a generator whose output confounder noise $N$ satisfies the causal relationship shown in Figure 3(a-2). Figure 3(a-1) illustrates our overall training pipeline, which mainly consists of three parts: a generator $G_\theta$ with learnable parameters $\theta$, a discriminator $D_\phi$ with learnable parameters $\phi$, and a pretrained classifier $f_\omega$ with fixed parameters $\omega$. The generator $G_\theta$ takes the private image $x$ as its input and generates a noise $G_\theta(x)$. The encrypted image $x + G_\theta(x)$ is obtained by combining the original image and noise, as highlighted in green in Figure 3(a-1). We emphasize that this noise as a confounder is better than the predefined class-wise patches and class-wise noises, because its generation relies on the original image, so that the two variables on path $N \to X$ have a very strong correlation (Figure 3(a-2)). Then $x + G_\theta(x)$ will be sent to the discriminator $D_\phi$, which is used to distinguish the generated data and the original data $x$. The training loss of the discriminator $D_\phi$ is

$$\mathcal{L}_\phi^{dis}(x) = \mathbb{E}_{x \sim \mathcal{P}_{data}} \log D_\phi(x) + \mathbb{E}_{x \sim \mathcal{P}_{data}} \log(1 - D_\phi(x + G_\theta(x))), \qquad (2)$$

where $\mathcal{P}_{data}$ represents the distribution of the original data. It can be found that the discriminator $D_\phi$ is a binary classifier that encourages the generated noise to be imperceptible.

Meanwhile, in order to ensure that the generated noise has the confounder property, the relationship $N \to Y$ in Figure 3(a-2) needs to be constructed. As highlighted in red in Figure 3(a-1), we achieve

**Algorithm 1:** Minibatch training of ConfounderGAN

---

**Input:** a generator $G_\theta$ with learnable parameters $\theta$, a discriminator $D_\phi$ with learnable
parameters $\phi$, a pretrained classifier $f_\omega$ with fixed parameters $\omega$, the training epoch is $M$,
the loss weight factor is $\alpha$.

**1 for** $m = 1, \cdots, M$ **do**

2     Sample minibatch of $B$ examples $\{x^{(1)}, \ldots, x^{(B)}\}$ with labels $\{t^{(1)}, \ldots, t^{(B)}\}$ from data
     distribution $\mathcal{P}_{data}$

3     Update the discriminator $D_\phi$ by ascending its gradient: $\nabla_\phi \frac{1}{m} \sum_{b=1}^{B} \mathcal{L}_\phi^{dis}(x^{(b)})$

4     Update the generator $G_\theta$ by descending its gradient:
     $\nabla_\theta \frac{1}{m} \sum_{b=1}^{B} \mathcal{L}_\theta^{confounder}(x^{(b)}, t^{(b)}) + \alpha \mathcal{L}_\theta^{hinge}(x^{(b)})$

---

this by classifying noise into the ground truth class $t$ of the original image. Thus training loss of the
confounder noise generator $G_\theta$ is

$$\mathcal{L}_\theta^{confounder}(x, t) = \mathbb{E}_{x \sim \mathcal{P}_{data}} \ell(f_\omega(G_\theta(x)), t), \tag{3}$$

where $\ell$ denotes the cross-entropy loss commonly used in classification tasks. Note that
$\ell(f_\omega(G_\theta(x)), t)$ in Equation (3) cannot be replaced with $\ell(f_\omega(x + G_\theta(x)), t)$, since $f_\omega(x)$ usu-
ally has a high classification accuracy for the original image $x$. This means that the loss $\ell(f_\omega(x), t)$
is low enough, then $\ell((f_\omega(x + G_\theta(x)), t)$ can be minimized as long as $G_\theta(x) = 0$.

To bound the magnitude of the noise and stabilize the GAN's training [13], we add a soft hinge loss
on the $L_2$ norm as

$$\mathcal{L}_\theta^{hinge}(x) = \mathbb{E}_{x \sim \mathcal{P}_{data}} \max(0, \|G_\theta(x)\|_2 - c), \tag{4}$$

where $c$ denotes a user-specified bound. Finally, we train the generator $G_\theta$ and discriminator $D_\phi$ by
solving a max-min game, as

$$\arg \min_{G_\theta} \max_{D_\phi} \mathcal{L}_\phi^{dis}(x) + \mathcal{L}_\theta^{confounder}(x, t) + \alpha \mathcal{L}_\theta^{hinge}(x), \tag{5}$$

where $\alpha$ controls the weight of $\mathcal{L}_\theta^{confounder}(x, t)$ and $\mathcal{L}_\theta^{hinge}(x)$ during generator training. See
Algorithm 1 for the pseudo-code of ConfounderGAN.

Once $G_\theta$ is trained, as illustrated in Figure 3(b), the data owner only needs to input the image into
the generator for one forward propagation to achieve the data encryption. If private data is used to
train the generator, the processed data is called in-distribution encryption data. Conversely, if private
data is NOT used to train the generator, the processed data is called out-of-distribution encryption
data. The latter setting is especially useful when the user has some new unlabeled data to encrypt, as
it does not need extra training. After this private data is encrypted, the data owner can upload it to
the World Wide Web. When the model trainer downloads these images, he/she cannot exploit useful
knowledge from these processed images, thus protecting the data privacy of the data owner.

## 5 Experiments

In this section, we first introduce our experimental setup. Then we demonstrate the effectiveness of
our proposed method in creating unlearnable examples under in-distribution and out-of-distribution
data encryption settings. Next, we conduct a series of transferability and stability of our confounder
noise to further illustrate the effectiveness and superiority of our method. Finally, we show some
visualizations to better understand our confounder noise.

### 5.1 Experimental setup

We select three natural object datasets and three medical datasets for algorithm evaluation, including
SVHN [22], CIFAR10 [16], ImageNet25 [5], BloodMNIST [45], Keratitis and ISIC [1]. See
Appendix F for more details on these datasets. We mainly focus on sample-wise noise since it can
be applied to various scenarios (*e.g.*, encrypting unlabeled out-of-distribution data). The proposed
method is compared with the error-minimizing noise (EMN) [12], which is the strongest baseline for

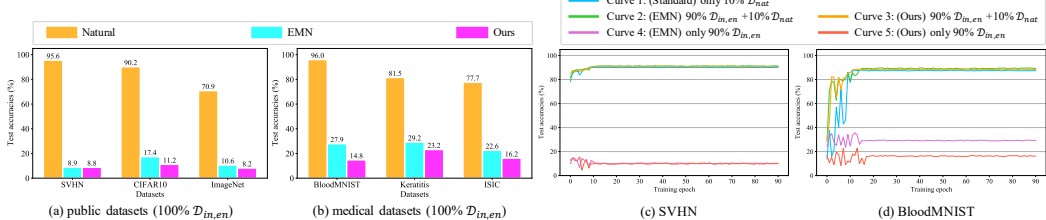

Figure 4: (a)∼(b) Test accuracies (%) of the model trained on 100% in-distribution encryption data $\mathcal{D}_{in,en}$ with ResNet18. (c)∼(d) Test accuracy curves on SVHN and BloodMNIST

.

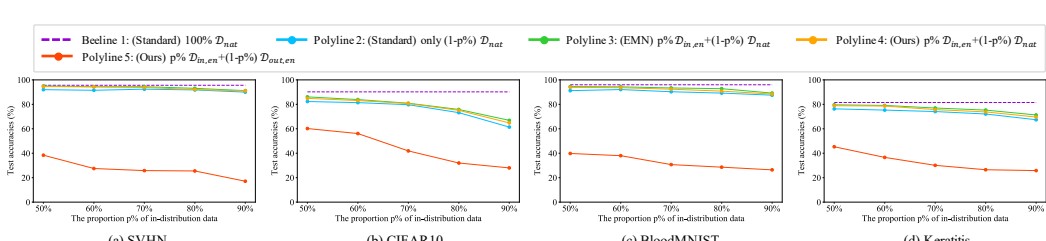

Figure 5: (a)∼(d) Effectiveness under different combinations of training data on various datasets with ResNet18 model: lower test accuracy indicates better effectiveness.

data privacy protection. We employ ResNet18 [10] as the source model or pretrained classifier for generating noise, and train ResNet18 on the unlearnable training sets. Referring to Huang et al. [12], the maximum perturbation is set to 8/255 in SVHN and CIFAR10, and 16/255 in other datasets, thus ensuring it is imperceptible to human observers. These settings are fixed for all experiments, unless otherwise explicitly stated. See Appendix G for more experimental details.

## 5.2 Effectiveness of ConfounderGAN

### 5.2.1 In-distribution data encryption

We first examine an extreme case, *if all data collected by the model trainer is in-distribution encryption data (i.e., $\mathcal{D}_{tr} = 100\%\mathcal{D}_{in,en}$), how does the model trained on this data perform?* The results are illustrated in Figure 4(a) and Figure 4(b), where 'natural' represents the original image. The corresponding test accuracies of EMN and our method demonstrate that the learnable knowledge of images is effectively suppressed. More importantly, all datasets processed by our method have the lowest accuracy, which proves the superiority of our method in data privacy protection.

However, in most situations, not all the training data collected by the model trainer are unlearnable examples. For example, only a certain number of web users have decided to use protection techniques but not all users. Thus, one question is naturally raised: *If the trainer uses $p\%$ in-distribution encryption data $\mathcal{D}_{in,en}$ and $(1 - p\%)$ natural data $\mathcal{D}_{nat}$ for model training, can these encryption data $\mathcal{D}_{in,en}$ still maintain the unlearnable property in this mixed dataset?* For validation, we train the model on only the natural proportion $(1 - p\%)\,\mathcal{D}_{nat}$ for comparison. The results are shown in Figure 5, where the X-axis represents the proportion $p\%$ of $\mathcal{D}_{in,en}$ to the original total dataset and the Y-axis represents the classification accuracy on the same test set $\mathcal{D}_{test}$. A quick glance at Polyline 3 and Polyline 4 tells us that the effectiveness of both EMN and our method drops quickly when the data are not made 100% unlearnable, because their accuracy is close to Beeline 1 (*i.e.*, 100% natural dataset). This means that the encryption effect can be approximately ignored even when the noise is applied to 50% of the data. However, we can also observe that Polyline 2 is close to Polyline 3 and Polyline 4, which may indicate that the high test accuracy in Polyline 3 and Polyline 4 is caused by $(1 - p\%)$ natural data $\mathcal{D}_{nat}$.

To illustrate this, we take $p = 90\%$ in SVHN and BloodMNIST as examples and plot their test accuracy curves during training as shown in Figure 4(c) and Figure 4(d). We additionally add Curve 4 and Curve 5 that only use $90\%$ in-distribution encryption data $\mathcal{D}_{in,en}$ for model training. The result of these two curves demonstrates that 90% $\mathcal{D}_{in,en}$ is still unlearnable and the test accuracy curve of our method is significantly lower than that of EMN on BloodMNIST, which is consistent with the

| Dataset | SVHN | | | CIFAR10 | | | ImageNet25 | | |
|---|---|---|---|---|---|---|---|---|---|
| | Natural | EMN | Ours | Natural | EMN | Ours | Natural | EMN | Ours |
| VGG11 | 95.0 | 32.6 | **25.4** | 86.2 | 19.1 | **16.9** | 62.6 | 34.0 | **26.5** |
| ResNet50 | 95.9 | 10.6 | **10.1** | 91.5 | 17.9 | **11.3** | 73.6 | 12.3 | **8.4** |
| DenseNet121 | 96.3 | 10.9 | **8.9** | 92.6 | 18.6 | **13.4** | 80.0 | 11.3 | **9.5** |
| Dataset | BloodMNIST | | | Keratitis | | | ISIC | | |
| | Natural | EMN | Ours | Natural | EMN | Ours | Natural | EMN | Ours |
| VGG11 | 96.3 | 35.4 | **29.7** | 79.8 | 55.4 | **35.6** | 72.6 | 45.6 | **18.7** |
| ResNet50 | 95.7 | 41.7 | **31.4** | 80.2 | 34.4 | **26.6** | 75.3 | 24.3 | **18.0** |
| DenseNet121 | 97.0 | 35.4 | **27.5** | 88.4 | 30.7 | **27.0** | 81.9 | 26.7 | **22.7** |

Table 1: Transferability results: the unlearnable noise is customized for ResNet18, while the evaluation is performed on VGG11, ResNet50, DenseNet121.

| Method | No augmentation | RHF | RC | RHF+RC |
|---|---|---|---|---|
| EMN | 17.4 | 17.9 | 18.0 | 18.3 |
| Ours | **11.2** | **11.9** | **14.8** | **14.9** |

| Method | CutOut | MixUp | CutMix | FA |
|---|---|---|---|---|
| EMN | 18.2 | 25.7 | 18.6 | **38.4** |
| Ours | **15.0** | **19.3** | **13.6** | 39.3 |

Table 2: Test accuracies (%) of the ResNet18 model trained on unlearnable CIFAR-10 with various data augmentation.

result of BloodMNIST in Figure 4(b) (100% $\mathcal{D}_{in,en}$). Meanwhile, Curve 1, Curve 2 and Curve 3 are almost the same, showing that the exploitable knowledge in Curve 2 and Curve 3 mainly comes from 10% natural data. Therefore, we conclude that both images encrypted by our method or EMN remain unlearnable property even as part of the dataset, and our method maintains its superiority to some extent. In addition, we also provide training gradient analysis on our encrypted images in the Appendix A to better understand the effectiveness of our method.

### 5.2.2 Out-of-distribution data encryption

In real-world scenarios, people sometimes want to quickly encrypt their instant photos with little computing resources. For example, a person takes a photo with a cell phone and plans to upload it to social media immediately. EMN is hard to deploy in this lightweight computing setting, as it needs multiple bi-level gradient optimizations to generate an effective noise. In contrast, in our ConfounderGAN, once the confounder noise generator $G_\theta$ is trained, it only needs to input any image into $G_\theta$ for one forward-propagation to achieve encryption, and no label information is required, which is ideal for the above practical scenario. However, since the generator has not seen these newly captured images during training, the effectiveness of the generated noise on these out-of-distribution images is unknown. Therefore, we try to explore the following question: *If the trainer uses $p\%$ in-distribution encryption data $\mathcal{D}_{in,en}$ and $(1 - p\%)$ out-of-distribution encryption data $\mathcal{D}_{out,en}$ for model training, can these out-of-distribution encryption data $\mathcal{D}_{out,en}$ effectively suppress the learnable information?* For brevity, we plot the experimental results to Polyline 5 in Figure 5. The generator for each point in Polyline 4 and Polyline 5 is trained with the same $p\%$ data, and performs in-distribution encryption on these data after training. The main difference between Polyline 4 and Polyline 5 is that the former does not encrypt the remaining $(1 - p\%)$ natural data, while the latter performs out-of-distribution encryption for these remaining data. It can find that the test accuracies of Polyline 5 are obviously lower than those of Polyline 4, which demonstrates that the out-of-distribution data encrypted by our method has significantly less learnable knowledge than the natural data $\mathcal{D}_{nat}$, thus proving the effectiveness of our method for out-of-distribution personal data. See Appendix E for more experiments.

### 5.3 A Series of Analytical Experiments about ConfounderGAN

For simplicity, the analytical experiments in this subsection are all conducted under the 100% in-distribution encryption setting (*i.e.*, 100% $\mathcal{D}_{in,en}$), unless otherwise stated.

### 5.3.1 Transferability analysis

Both EMN and our ConfounderGAN rely on a pretrained classifier $f$ to generate effective unlearnable noise. Specifically, EMN uses this classifier as a source model to compute the gradient of error-

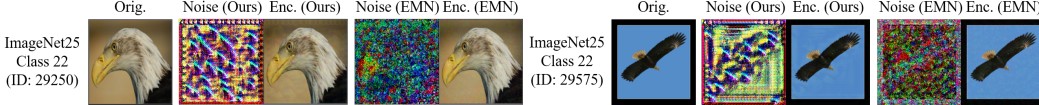

Figure 6: Visualization: Orig. represents the original image, Enc. represents the encrypted image.

minimizing noise, and our ConfounderGAN needs this classifier to learn the correlation between the confounder noise and the label. Therefore, it can be considered that unlearnable noise is customized for this classifier $f$. If the pretrained classifier $f$ used by the data owner and the commercial model $h$ trained by the unauthorized company share the same backbone, the protection performance should be optimal and the above experiments follow this setting (*i.e.*, both are ResNet18). However, in practical scenarios, the data owner cannot predict which model $h$ the trainer will choose as the backbone. A natural concern is raised: *If the architecture of classifier $f$ and model $h$ are different, can the unlearnable property of the noise trained on classifier $f$ be transferable to the model $h$?* Table 1 shows the results of the transferability under in-distribution encryption setting (*i.e.*, 100% $\mathcal{D}_{in,en}$), where the classifier $f$ employs ResNet18 and the model $h$ selects VGG11 [34], ResNet50 and DenseNet121 [11]. The results show our method outperforms EMN on all datasets and evaluated models, especially for VGG11, indicating that our confounder noise is more practical and effective.

### 5.3.2 Stability analysis

Model trainers are likely to use some data augmentation techniques to improve their model performance during the training phase, so a question is introduced: *Can noise still work well with different data augmentations?* To answer this question, we select 7 commonly used and advanced data augmentation techniques for evaluation on CIFAR10: Random Horizontal Flip (RHF), Random Crop (RC), Combination of Horizontal Flip and Crop (RHF+RC), CutOut [7], MixUp [47], CutMix [46], Fast Autoaugment (FA) [19]. See Appendix G.6 for detailed experimental setup. The results are shown in Table 2. It can be observed that the noise still maintains high effectiveness in most settings. Meanwhile, the test accuracies of our method are also lower than those of EMN in all cases, which further proves that our method is better than EMN. In addition, we also investigate the stability of ConfounderGAN under early stopping and adaptive setting. See Appendix C and Appendix D for more details.

### 5.4 Visualizations

As shown in Figure 6, we randomly select two images from the ImageNet25 for visualization. Since noises are all limited to a small range, all noises are scaled to the interval $[0, 255]$ according to their maximum and minimum values. We can obverse that the noises of EMN are small disordered dots in most cases, while the noises of our method show some high-level representations. *e.g.*, feather-like textures can be observed. These high-level representations may lead to the better transferability of our method (as mentioned in Section 5.3.1), since different backbones may share some common representations. See Appendix B for more visualizations.

## 6 Conclusion

We focus on the image data encryption task, its goal is to use small and invisible noise to prevent personal data from being freely exploited by deep neural networks. By leveraging confounder property, we propose a confounder-based encryption framework for image data. *i.e.*, introduce a confounder that is strongly correlated with both images and labels in the dataset, so that the model trained on this dataset can only learn the spurious correlations between images and labels. Based on this framework, we develop confounderGAN, a novel image data encryption method that covers various practical scenarios. Empirically, we demonstrate that our method outperforms the state-of-the-art algorithm in the in-distribution data encryption and can be effectively applied to out-of-distribution data encryption. Moreover, we also verify that the transferability and robustness of our confounder noise. We believe this paper provides new insight into data privacy protection and could have a broad impact on both the public and the deep learning community.

# 7 Acknowledgements

This work was supported in part by National Natural Science Foundation of China (No. 62006207, No. 62037001), the Starry Night Science Fund of Zhejiang University Shanghai Institute for Advanced Study (SN-ZJU-SIAS-0010), Key Laboratory for Corneal Diseases Research of Zhejiang Province, Project by Shanghai AI Laboratory (P22KS00111), Program of Zhejiang Province Science and Technology (2022C01044), the Fundamental Research Funds for the Central Universities (226-2022-00142, 226-2022-00051), Zhejiang Province Natural Science Foundation (No. LQ21F020020) and National Natural Science Foundation of China (Grant No. U20A20387).

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
