# A Gradient analysis

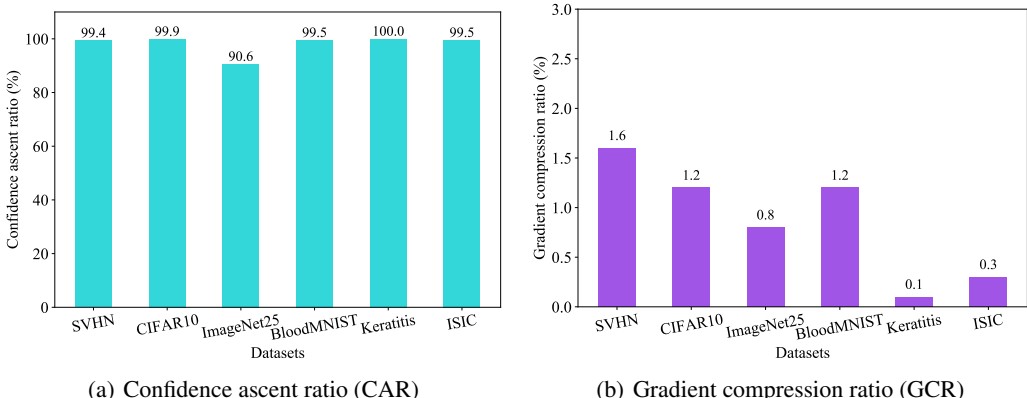

(a) Confidence ascent ratio (CAR)

(b) Gradient compression ratio (GCR)

Figure 7: Confidence ascent ratio (CAR) and Gradient compression ratio (GCR) on various datasets.

To better understand why our generated confounder noise can make the data unlearnable, we can also gain some insights according to optimization gradient. Empirically, if one image provides a large gradient in a backpropagation, this image has a lot of learnable knowledge, and vice versa. Thus a natural question is: *How do the confidence and optimization gradients produced by our encrypted dataset change relative to those of the original dataset during model training?* We propose two statistical metrics for validation: Confidence Ascent Ratio (CAR) and Gradient Compression Ratio (GCR). Specifically, suppose that the training model is $h$ with the parameters of the last layer (classifier) $W$, the number of images in the training set $\mathcal{D}_{tr}$ is $n$. The confidence of the original image $x$ corresponding to ground truth class $t$ is $h(x)_t$, and this confidence of the encrypted image is $x + G_\theta(x)$ is $h(x + G_\theta(x))_t$, then CAR is defined as

$$\text{CAR} = \frac{\sum_{x \in \mathcal{D}_{tr}}^{\mathcal{D}_{tr}} \mathbb{1}((h(x + G_\theta(x))_t - h(x)_t) > 0)}{n} \,, \tag{6}$$

where $\mathbb{1}(\cdot)$ is a indicator function. GCR is defined as

$$\text{GCR} = \frac{\sum_{x \in \mathcal{D}_{tr}}^{\mathcal{D}_{tr}} (\|\nabla_W \ell(x + G_\theta(x), t)\|_2 / \|\nabla_W \ell(x, t)\|_2)}{n} \,. \tag{7}$$

where $\| \cdot \|_2$ represents the $L_2$ norm, $\nabla_W \ell(\cdot)$ represents the gradient of the given input w.r.t. the classifier.

Figure 7 demonstrates CAR and GCR in all datasets, where the checkpoint of model $h$ is randomly selected during the training phase. From the result of Figure 7(a), we can find that CAR exceeds 90% in all datasets and even reaches 100% in Keratitis. This means that most images can be correctly classified by the model $h$ after adding our confounder noise, so that the model 'thinks' that these encrypted images have nothing to learn. This phenomenon can be understood because the confounder noise and the label have a strong correlation in ConfounderGAN. Further, we can observe that GCR is less than 2% for all datasets in Figure 7(b). That is, the backpropagation gradient provided by the encrypted image $x + G_\theta(x)$ to the deep neural network is less than 2% of that provided by the original image $x$, which fully shows the exploitable knowledge in $x + G_\theta(x)$ is greatly suppressed compared to $x$.

# B    Visualization

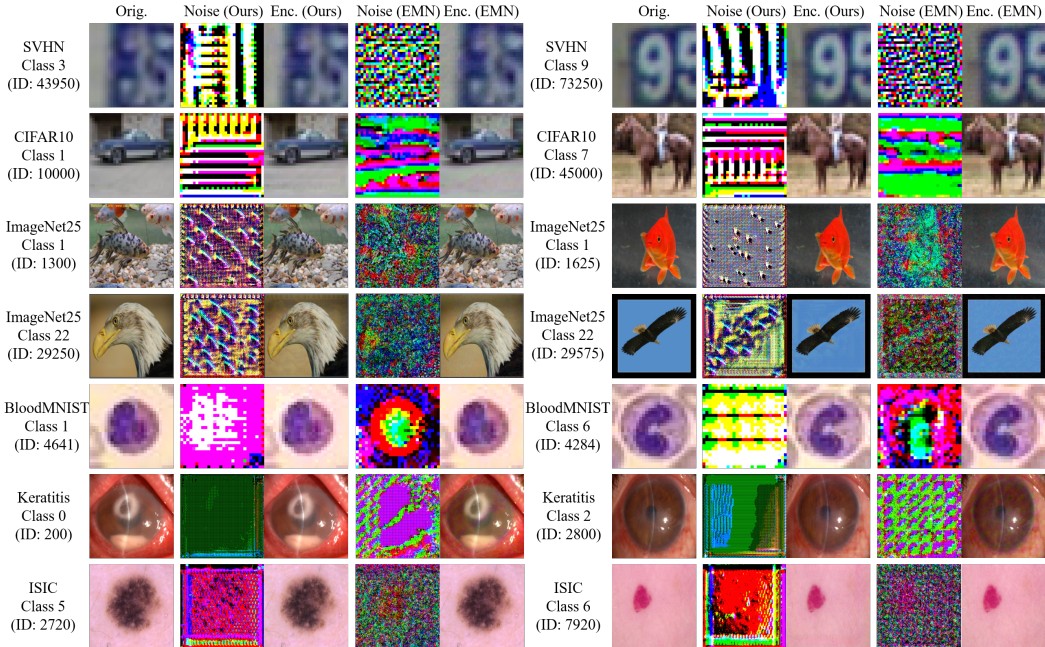

Figure 8: Visualization: Orig. represents the original image, Enc. represents the encrypted image.

# C    Stability analysis about early stopping

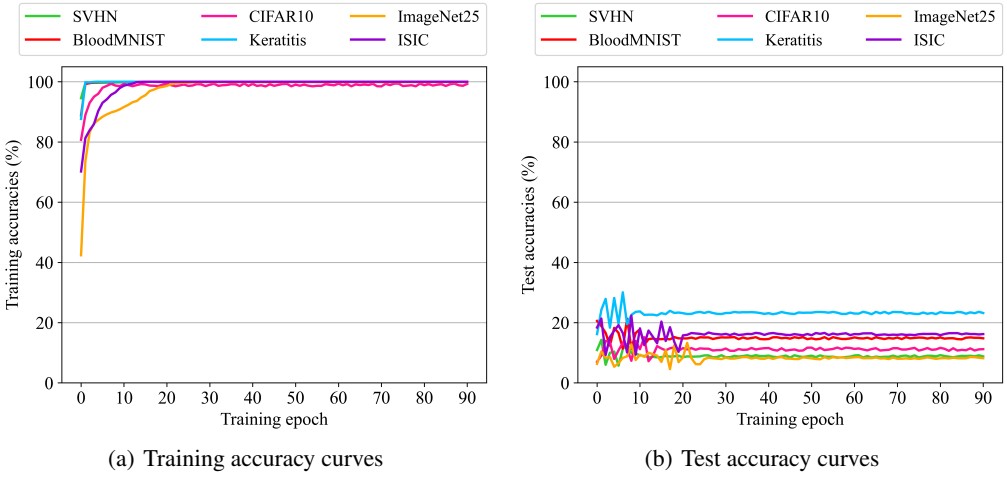

(a) Training accuracy curves

(b) Test accuracy curves

Figure 9: Training and test accuracy curves on different datasets during model training.

Huang et al. [12] find that if the model is trained on the dataset with predefined class-wise noises, it can achieve satisfactory test accuracy in the early stage of training. This means that the model trainer can use early stopping to circumvent the protection of predefined class-wise noises. Then they prove that their proposed EMN does not suffer from the above phenomenon. We are also curious: *Is the noise generated by our ConfounderGAN effective for the entire epoch of training?* Figure 9 shows the accuracy curves of our method during the training epoch. It reveals that the model always has low test accuracy at all stages of training, so the early stopping trick cannot circumvent our confounder noise.

# D    Stability analysis about adaptive setting

In this section, we first conduct the experiment to investigate the effectiveness of our method under the adaptive setting proposed by [28]. Then we give a detailed discussion about this setting.

To better understand this adaptive setting, we first illustrate the assumption on the data owner's capability and the model trainer's capability under this setting:

**Assumption on data owner's capability:** A data owner processes some natural images $\mathcal{D}_{nat}$ into encrypted images $\mathcal{D}_{en}$ via ConfounderGAN and uploads them to social media.

**Assumption on model trainer's capability:** A model trainer knows that these images $\mathcal{D}_{en}$ have been processed by ConfounderGAN and can directly access the generator of ConfounderGAN. The model trainer wishes to train a denoiser against the noise generated by the ConfounderGAN. However, the trainer cannot obtain the original images $\mathcal{D}_{nat}$ corresponding to the encrypted images $\mathcal{D}_{en}$, otherwise he/she can directly use these original images $\mathcal{D}_{nat}$ to train the model. Therefore, in order to denoise the encrypted images $\mathcal{D}_{en}$, the model trainer needs to do the following steps: 1) collect additional natural images $\mathcal{D}'_{nat}$. 2) feed the surrogate images $\mathcal{D}'_{nat}$ into the generator of ConfounderGAN to build the encrypted images $\mathcal{D}'_{en}$. 3) use $\mathcal{D}'_{nat}$ and $\mathcal{D}'_{en}$ to train a denoiser. 4) remove the noise of the encrypted images $\mathcal{D}_{en}$ by the trained denoiser.

We conduct the experiment on CIFAR10 to investigate whether the adaptive denoiser can completely remove the effect of the encrypted noises. In practice, we divide the training set of CIFAR10 into two equally as $\mathcal{D}_{nat}$ and $\mathcal{D}'_{nat}$, and then use the above steps to obtain the denoised images, where the training of the denoiser follows DnCNN [48]. The experimental results are shown in the table below.

| Training dataset | Natural images | Denoised images | Encrypted images |
|---|---|---|---|
| Test accuracy | 87.4% | 77.9% | 11.9% |

Table 3: Test accuracies (%) of the model trained on different datasets.

The result shows that although the denoiser can resist our encryption method to a certain extent, the model's performance can still be compromised by our confounder noises, which shows that our method retains its effectiveness under the adaptive setting.

Meanwhile, for the adaptive setting proposed by Radiya-Dixit et al. [28], we believe there are two important points that need to be clarified.

**1) We believe that this adaptive setting has unbalanced assumptions about the strength of the data owner's capability and model trainer's capability, leaving the data owner (or crypto tool designer) on the weaker side.** Specifically, Radiya-Dixit et al. [28] assume that the model trainer has full access to the encryption tool, while the crypto tool designer has no knowledge of the model trainer's decryption method. For example, in our paper, ConfounderGAN's designer does not know that the model trainer will use a denoiser to remove encryption noises. However, if we know this information in advance, we might be able to introduce the knowledge of the denoiser into the training process of the ConfounderGAN, making it robust to the denoiser. An intuitive idea is to change the existing training architecture from 'original image -> generator -> confounder noise -> pretrain classifier' to 'original image -> generator -> confounder noise -> **pretrain denoiser** -> pretrain classifier', so that the confounder property can be preserved even if the generated noises encounter a denoiser in the future. Of course, this solution is very rudimentary. We will explore the optimal solution in future work, thus giving the data owner (or crypto tool designer) an edge in the arms race with the model trainer.

**2) Since data owners usually don't reveal which encryption tool they use, we believe that the non-adaptive setting may be more practical than the adaptive setting in real-world scenarios.** Specifically, Radiya-Dixit et al. [28] believe that the adaptive setting is practical in the real-world, and they give the following argument: encryption tools are usually publicly accessible applications, thus model trainers can adaptively train a feature extractor that resists these encryption noises. We agree that encryption tools are generally publicly accessible, but disagree that model trainers can adaptively train feature extractors. This is because multiple encryption methods will be proposed in the future. When data owners publish encrypted data, they won't reveal the encryption tools they use in most scenarios. Thus it is difficult for the model trainers to determine which encryption method should be

used when adaptively training decryption feature extractors. In fact, referring to the community of adversarial examples [14, 15], one encryption method can derive multiple instantiations by modifying the encryption constraints. For example, data owners can replace small pixel-wise perturbation with watermark [14], color channel perturbation [15], *etc.*, according to their preferences. As long as the data owner does not expose the information of the encryption tool, the model trainer cannot decrypt it in an adaptive manner. Based on these analyses, we believe that the non-adaptive setting may be more practical than the adaptive setting in real-world scenarios. Note that this does not mean that the adaptive setting is unnecessary, we realize that it is important to design an encryption tool that strictly satisfies the Kerckhoffs's principle [25]. In future work, we will further improve ConfounderGAN so that it can work optimally under this principle.

# E  Comparing CounfounderGAN with DeepConfuse under out-of-distribution encryption

We notice that DeepConfuse [8] can also be applied to out-of-distribution (OOD) data encryption. Therefore, we compare the effectiveness of this method in the OOD setting with our method. The experiment is conducted on CIFAR10 and the evaluation settings are consistent with Figure 6(b) of the manuscript. The experimental results are as follows:

| Method | p% $\mathcal{D}_{in,en}$ + (1-p%) $\mathcal{D}_{out,en}$ | | | | |
|---|---|---|---|---|---|
| | p=50% | p=60% | p=70% | p=80% | p=90% |
| DeepConfuse | 71.3% | 61.6% | 50.2% | 45.8% | 36.3% |
| Ours | 60.2% | 56.1% | 41.9% | 32.0% | 28.0% |

Table 4: Test accuracies (%) under different combinations of training data.

We can find that the dataset processed by our ConfounderGAN can obtain lower test accuracy, indicating that our method outperforms DeepConfuse for out-of-distribution encryption.

# F  Datasets

We show some key information about each dataset to better understand our experiments.

**SVHN**[2]**.** SVHN is a digit dataset, containing numbers 0 to 9. This dataset is collected from house numbers in Google Street View images with low resolution, which consists of 73,257 training images and 26,032 test images in 10 classes. The number of each class in the training and test sets is unbalanced. All images are $3 \times 32 \times 32$ three-channel color images.

**CIFAR10**[3]**.** CIFAR10 are labeled subsets of the 80 million tiny images dataset. The latter is automatically downloaded from the Internet through a crawler script, and this unauthorized data collection matches the scenario assumed in this paper. CIFAR10 consists of 50,000 training images and 10,000 test images in 10 classes, classes, with 5,000 and 1,000 images per class. All images are $3 \times 32 \times 32$ three-channel color images.

**ImageNet25**[4]**.** ImageNet25 is a subset of the ImageNet dataset (the first 25 classes). The experiments on this dataset are to confirm the effectiveness of the method on high-resolution images. It consists of 32,500 training images and 1,250 test images, with 1,250 and 50 images per class. All images are $3 \times 224 \times 224$ three-channel color images.

**BloodMNIST**[5]**.** BloodMNIST is based on a dataset of individual normal cells, captured from individuals without infection, hematologic or oncologic disease. It is organized into 8 classes and consists of 11,959 training images and 3,421 test images. The number of each class in the training and test sets is unbalanced. All images are $3 \times 28 \times 28$ three-channel color images.

---

[2] http://ufldl.stanford.edu/housenumbers/

[3] https://www.cs.toronto.edu/~kriz/cifar.html

[4] https://www.image-net.org/

[5] https://medmnist.com/

**Keratitis.** This dataset is collected at our local hospital to evaluate the effectiveness of the method in real-world scenarios. It consists of 4,047 training images and 581 test images in 4 classes. The number of each class in the training and test sets is unbalanced. All images are $3 \times 224 \times 224$ three-channel color images.

**ISIC[6].** ISIC is a high-resolution medical datasets, which is collected from leading international clinical centers. This dataset consists of 8,005 training images and 2,010 test images in 7 classes. The number of each class in the training and test sets is unbalanced. All images are $3 \times 224 \times 224$ three-channel color images.

## G    Experimental setup

### G.1    Hardware

In all experiments, the GPU is NVIDIA GTX 1080Ti and the CPU is Intel(R) Xeon(R) E5-2678 v3 @ 2.50GHz.

### G.2    Experimental setup for toy experiments

In the class-wise patches experiment, we select the first 25 classes from ImageNet to construct a new dataset ImageNet25 for evaluation. In the training set, each $224 \times 224$ image adds a $32 \times 32$ patch by class in the lower right corner. Since images of ImageNet are color images, the added patches are also three-channel color images. To simplify, we design the value of each patch by channel. *e.g.*, the patch for Class 0 is represented as (220,0,0), which means that all the pixels of the first channel are 220 and the other channels are all 0. Then we list the patch for all classes: (220,0,0),(230,0,0),(240,0,0),(250,0,0),(0,220,0),(0,230,0),(0,240,0),(0,250,0),(0,0,220),(0,0,230),(0,0,240),(0,0,250),(220,220,0),(230,230,0),(240,240,0),(250,250,0),(220,0,220),(230,0,230),(240,0,240),(250,0,250),(220,0,220),(230,0,230),(240,0,240),(250,0,250),(250,250,250).

We use ResNet18 as the backbone. The hyperparameters for model training are listed as follows: the optimizer is SGD, momentum is 0.9, initial learning rate is 0.025, the learning rate scheduler is cosine scheduler without the restart, training epoch is 90.

### G.3    Experimental setup for ConfounderGAN

ConfounderGAN contains three neural networks: a generator, a discriminator and a pretrained classifier. The generator consists of a 4-layer convolutional neural network and a 4-layer deconvolutional neural network. The discriminator consists of a 4-layer convolutional neural network and a binary classifier. The pretrained classifier is ResNet18. The input and output layers of the generator and pretrained classifier are adjusted according to the image size and class number of each dataset. The classifiers for all datasets are pretrained on their training set, except for ImageNet where the pretrained model is downloaded directly from the Pytorch official website.

The training of ConfounderGAN consists of generator training and discriminator training. The hyperparameters for generator training is listed as follows: the optimizer is SGD, the momentum is 0.9, the initial learning rate is 0.025, the learning rate scheduler is cosine scheduler without the restart, the loss weight factor is 0.001, the training epoch is 200(SVHN,CIFAR10,Keratitis)/5000(ImageNet25)/100(BloodMNIST)/400(ISIC). The training hyperparameters of the discriminator refer to those of the generator, except that the learning rate is set to a constant 0.025.

### G.4    Experimental setup for EMN

EMN requires a source model as a surrogate model to compute the noise gradient. Referring to the original paper, the backbone of this model architecture is ResNet18.

EMN is a two-level optimization method. The inner loop generates noise, and its hyperparameters are as follows: the optimizer is SGD, the learning rate is 0.003, the number of iterations is 20. The outer loop updates the source model with the following hyperparameters: the optimizer is SGD, the

---

[6]`https://challenge2018.isic-archive.com/`

learning rate is 0.003, the number of iterations is 10. The stopping condition for training is that the error of the source model on the training set is 0.01.

### G.5 Experimental setup for evaluation

We need to train a model $h$ on the unlearnable dataset to prove the effectiveness of the encryption algorithm. In our experiments, all datasets share a set of hyperparameters for training model $h$: the optimizer is SGD, the momentum is 0.9, the initial learning rate is 0.025, the learning rate scheduler is cosine scheduler without the restart, the training epoch is 90.

### G.6 Experimental setup for data augmentations

For RC, we set the padding length to 4 pixels. For CutOut, we set the cutout length to 16 pixels. For MixUp, we apply linear mixup of random pairs of training examples and their labels during the training process. For CutMix, we apply linear mixup on the cutout region. For FA, we use the fixed augmentation policy, which consists of change contrast, brightness, sharpness, rotations and cutout.