# OpenReview forum: "ConfounderGAN: Protecting Image Data Privacy with Causal Confounder"
_NeurIPS.cc/2022/Conference — NeurIPS 2022 Accept_

### Official Review · Reviewer_rw45 · 2022-07-10

**Rating:** 8
**Confidence:** 4
**Soundness:** 3 good
**Presentation:** 3 good
**Contribution:** 3 good

**Summary:**

This paper proposes a GAN that makes personal image data unlearnable by DL methods for data protection. The authors utilize the confounder property present in the noise produced by the generator. This property builds spurious correlations between images and labels, disallowing the model to learn the correct mappings. The discriminator is used to ensure that this generated noise is undetectable. The authors conduct experiments using six image classification datasets, 3 of which are natural object datasets and 3 are medical datasets. Specifically, a confounder based framework has been proposed for image data encryption.

**Questions:**

Section 2, Related work reviews data privacy, data poisoning and causal confounder in DL. However, only data privacy and confounder in deep learning have been discussed. Has data poisoning been combined with data privacy? There was no clear demarcation in the discussion of the two concepts under Data Privacy.
Section 5.2.1 could be better structured for ease of reading.

**Limitations:**

There are very few possible negative societal impacts, that are not straightforward in nature. The authors have, however, discussed the positive impact, which is data privacy in DL.

**Strengths And Weaknesses:**

The paper has been very well written, introduction to concepts have been well laid out and the structure of the paper is clear. The authors have done a thorough study of the past works in this particular domain. There are, however, some typos throughout the paper. For example: Line 12 -> "thereby, remaining the normal utility", Line 14 -> "The experiments are conducted in six image classification datasets, including three natural object datasets and three medical datasets" (this reads slightly ambiguous so a better choice of word for including might be consisting/comprising), Line 237 -> sentence capitalization.

---

> ### Author Response · Authors · 2022-08-01
> **Answer weaknesses and question 1-2 (1/1)**
>
> Thanks for the reviewer’s constructive feedback. We answer the questions in the Weaknesses and Questions:
>
> **Weaknesses: There are some typos throughout the paper.**
>
> **A.** Thank you for your careful review. For the typos in Line 12, Line 14 and Line 237 in the original manuscript, we have corrected them in the revised version.
>
> **Q1. Section 2, Related work reviews data privacy, data poisoning and causal confounder in DL. However, only data privacy and confounder in deep learning have been discussed. Has data poisoning been combined with data privacy? There was no clear demarcation in the discussion of the two concepts under Data Privacy.**
>
> **A1.** The relation between our method and traditional data poisoning is discussed in the related work of the revised manuscript. Note that some privacy protection methods and traditional data poisoning methods both deal with the training-time dataset. However, traditional data poisoning methods usually carry the purpose of malicious attack, while the motivations of our work, EMN [1], and Fawkes [2] are to protect the privacy of data owners. Therefore, we classify the latter into Data Privacy in the related work. The added details of data poisoning are as follows:
>
> **Data poisoning:** The goal of traditional data poisoning is to reduce the test accuracy of the model by modifying the training set. Biggio et al. [3] first introduce this type of attack in support vector machines [4]. Then Muñoz-González et al. [5] propose a deep learning version by poisoning the most representative samples in the training examples. Although data poisoning attacks seem to share the same goal as ours, these methods have a limited impact on DNNs and are not suitable for data protection tasks. e.g., the model trained on poisoned examples can still achieve acceptable performance [5], and the modified image can be easily distinguished from the original one [6]. In addition, the backdoor attack is another type of attack that poisons training data with a trigger pattern [7,8,9], but this attack does not prevent the model from learning useful knowledge in the natural data. Therefore, traditional data poisoning methods cannot be used for data protection, while our proposed method can produce unlearnable examples with imperceptible noise.
>
> [1] Unlearnable Examples: Making Personal Data Unexploitable. Huang et al. ICLR, 2021.
> [2] Fawkes: Protecting privacy against unauthorized deep learning models. Shan et al. USENIX Security. 2020.
> [3] Poisoning attacks against support vector machines. Biggio et al. ICML, 2012.
> [4] Support vector machines. Hearst et al. IEEE Intelligent Systems and their applications, 1998.
> [5] Towards poisoning of deep learning algorithms with back-gradient optimization. Muñoz-González et al. ACM workshop on artificial intelligence and security, 2017.
> [6] Generative poisoning attack method against neural networks. Yang et al. arXiv preprint arXiv:1703.01340, 2017.
> [7] Targeted backdoor attacks on deep learning systems using data poisoning. Chen et al. arXiv preprint arXiv:1712.05526, 2017.
> [8] Poison frogs! targeted clean-label poisoning attacks on neural networks. Shafahi et al. NeurIPS, 2018.
> [9] Reflection backdoor: A natural backdoor attack on deep neural networks. Liu et al. ECCV, 2020.
>
> **Q2. Section 5.2.1 could be better structured for ease of reading.**
>
> **A2.** In the revised manuscript, we have reorganized the paragraphs of Section 5.2.1 to make their structure clearer.

---

### Official Review · Reviewer_Bo5j · 2022-07-11

**Rating:** 7
**Confidence:** 5
**Soundness:** 3 good
**Presentation:** 3 good
**Contribution:** 3 good

**Summary:**

This paper proposed using GAN to produce confounder noise for unlearnable examples. The proposed method address an important issue that personal data is being used for unauthorized machine learning training. Unlike existing methods that require bi-level optimizations with multiple backward passes, the proposed method can generate confounder noise in a forward pass after training, making it very practical in a real-world application. Empirically, the proposed method outperforms existing methods.


**Questions:**

Please address the questions in the Strengths And Weaknesses section.



**Limitations:**

Please address the potential limitations in the Strengths And Weaknesses section.

**Strengths And Weaknesses:**

Strengths
- Well-motivated method for efficiently generating unlearnable examples, and the context of unauthorized machine learning training is well explained.
- The proposed method is efficient and technically sound. Existing works rely on optimizations that may not be practical for a user to generate unlearnable examples on the fly. Using GAN, the unlearnable version of the image can be generated in a forward pass, which improves usability in a practical setting.
- Comprehensive empirical evaluations of different datasets and models are appreciated. Results demonstrated the proposed method consistently outperforms existing methods.

---
Weaknesses/Limitations:
- For experiments on the different proportions of unlearnable examples, it does not make sense to compare Polyline 5 with others. Polyline 5 is still regarded as the 100% unlearnable case. It would be interesting to see $ D_{out,en} $ with $D_{nat}$.
- Legends on Fig 5 (c-d) color is not clear for EMN v.s. the proposed method for only $ D_{in,en} $
- Line 215, "When the model trainer downloads these images..." I believe the goal of unlearnable examples is to make the model unable to predict the protected classes/users rather than high-performing models.
- Line 237 "we first" -> "We first"
- Once the data is released, the defender may not modifies the data anymore, and the model trainer can retroactively apply new models/methods [1]. An adaptive case should be carefully examined.
- Comparison with DeepConfuse [2], which also able to generate unleranable samples for $ D_{out,en} $

[1] Data Poisoning Won’t Save You From Facial Recognition, ICML 2021 Workshop AML
[2] Learning to Confuse: Generating Training Time Adversarial Data with Auto-Encoder. NeurIPS 2019


---
After the author's response, I increased my rating score to 7.

A detailed analysis of the adaptive method to reverse the original image is well explained and discusses the potential limitation of the proposed method. Based on the author's response, in practice, the owner should keep the parameters of ConfounderGAN private to prevent the model trainer reverse the unlearnable data.

---

> ### Author Response · Authors · 2022-08-01
> **Answer question 1-4 (1/1)**
>
> We thank the reviewer’s constructive feedback and answer each question below:
>
> **Q1. 1) For experiments on the different proportions of unlearnable examples, it does not make sense to compare Polyline 5 with others. Polyline 5 is still regarded as the 100% unlearnable case. 2) It would be interesting to see $D_{out,en}$ with $D_{nat}$.**
>
> **A1. 1)** Each point of Polyline 5 uses only p% of the dataset for training the ConfounderGAN. During the evaluation phase, the confounder property of our method needs to generalize to (1-p)% out-of-distribution (OOD) data. If ConfounderGAN cannot achieve this, then Polyline 5 (p% $D_{in,en}$ + (1-p%) $D_{out,en}$) and Polyline 4 (p% $D_{in,en}$ + (1-p%) $D_{nat}$) will be close. The results show that Polyline 5 is significantly lower than Polyline 4, indicating that our ConfounderGAN is still effective for OOD data. We elaborate on the above statement in Section 5.2.2 of the manuscript.
>
> **A1. 2)** In this paper, our purpose is consistent with Huang et al. [1], i.e., use ConfounderGAN to encrypt the original images as unlearnable examples, so that the model trainer cannot exploit useful knowledge from these processed images. When the training dataset is a mixed dataset consisting of encrypted images $D_{en}$ and natural images $D_{nat}$, it is an interesting question whether these encrypted images $D_{en}$ are still unlearnable. We explore this problem in Figure 4(c)~4(d) of the manuscript (the corresponding description is Line 274 - Line 283). Taking bloodMNIST (Figure 4(d)) as an example.
> A quick glance at Curve 2 (EMN: 90% $D_{en}$+10% $D_{nat}$) and Curve 3 (Ours: 90% $D_{en}$+10% $D_{nat}$) tells us that the effectiveness of both EMN and our method drops quickly when the data are not made 100% unlearnable (encryption).
> However, the result of Curve 4 (EMN: only 90% $D_{en}$) and Curve 5 (Ours: only 90% $D_{en}$) demonstrates that the 90% $D_{en}$ is still unlearnable, and our method is more efficient.
> Meanwhile, Curve 1 (only 10% $D_{nat}$), Curve 2 and Curve 3 are almost the same, showing that the exploitable knowledge in Curve 2 and Curve 3 mainly comes from 10% natural data.
> Therefore, we conclude that both images encrypted by our method or EMN remain unlearnable property even as part of the dataset, and our method maintains its superiority to some extent.
> Although this investigation is conducted under in-distribution encryption, it can naturally generalize to the out-of-distribution setting.
>
> [1] Unlearnable Examples: Making Personal Data Unexploitable. Huang et al. ICLR, 2021.
>
> **Q2. Legends on Fig 5 (c-d) color is not clear for EMN v.s. the proposed method for only $\mathcal{D}_{in}$.**
>
> **A2.** Thank you for your careful review, we've updated the color of Curve 5 in the legend to make it easier to distinguish from Curve 4.
>
> **Q3. Line 215, "When the model trainer downloads these images..." I believe the goal of unlearnable examples is to make the model unable to predict the protected classes/users rather than high-performing models.**
>
> **A3.** Thank you for your careful review. In the revised manuscript (Line 233), we have revised this description as follows: When the model trainers download these images, they cannot exploit useful knowledge from these processed images, thus protecting the data privacy of the data owner.
>
> **Q4. Line 237 "we first" -> "We first"**
>
> **A4.** Thank you for your careful review. We have corrected this typo in the revised manuscript.

---

### Official Review · Reviewer_oeZM · 2022-07-16

**Rating:** 6
**Confidence:** 5
**Soundness:** 2 fair
**Presentation:** 2 fair
**Contribution:** 2 fair

**Summary:**

This paper proposes ConfounderGAN, a GAN whose generator can be used to create a noise to an image to make it *unlearnable*, by creating a spurious correlation between the image and the label. The proposed approach has been evaluated on several image classification tasks and the results show it can help reduce the accuracy of a model trained on noisy data in the *non-adaptive* setting.

**Questions:**


- Please consider discussing the relation between this solution and data poisoning in more detail.
- I suggest, using "perturbed" images instead of "encrypted" images
- The current discussion on how attackers can bypass the defence is a bit limited. What happens when some users have both their perturbed and un-perturbed data online? Since ConfounderGAN can be obtained by any user, the attacker can also use it to train a denoiser. Overall, more discussion on the adaptive setting could help.

**Limitations:**

Nothing to report.


**Strengths And Weaknesses:**


**Strengths**

The paper tackles an important issue. The results seem promising. However, several issues need to be addressed to make the paper more convincing.

**Weaknesses**

- The proposed approach seems a lot like a data poisoning attack. However, discussions on data poisoning and its relation to this solution are missing. Instead, terms like encryption are used, which can be misleading.
- The paper did not discuss the asymmetry between users and attackers as discussed in recent literature (e.g., [a]), which may give a false sense of security to users as these types of countermeasures have been proven to be ineffective.

[a]- Data Poisoning Won't Save You From Facial Recognition. (Radiya-Dixit et al., 2021)

---

> ### Author Response · Authors · 2022-08-01
> **Answer question 1 (1/1)**
>
> Thanks for the reviewer’s constructive feedback. We find that the comments of Weaknesses and Questions are consistent, thus we mainly focus on solving the three questions below.
>
> **Q1. Please consider discussing the relation between this solution and data poisoning in more detail.**
>
> **A1.** The relation between our method and traditional data poisoning is discussed in the related work of the revised manuscript. Note that some privacy protection methods and traditional data poisoning methods both deal with the training-time dataset. However, traditional data poisoning methods usually carry the purpose of malicious attack, while the motivations of our work, EMN [1], and Fawkes [2] are to protect the privacy of data owners. Therefore, we classify the latter into Data Privacy in the related work. The added details of data poisoning are as follows:
>
> **Data poisoning:** The goal of traditional data poisoning is to reduce the test accuracy of the model by modifying the training set. Biggio et al. [3] first introduce this type of attack in support vector machines [4]. Then Muñoz-González et al. [5] propose a deep learning version by poisoning the most representative samples in the training examples. Although data poisoning attacks seem to share the same goal as ours, these methods have a limited impact on DNNs and are not suitable for data protection tasks. e.g., the model trained on poisoned examples can still achieve acceptable performance [5], and the modified image can be easily distinguished from the original one [6]. In addition, the backdoor attack is another type of attack that poisons training data with a trigger pattern [7,8,9], but this attack does not prevent the model from learning useful knowledge in the natural data. Therefore, traditional data poisoning methods cannot be used for data protection, while our proposed method can produce unlearnable examples with imperceptible noise.
>
> [1] Unlearnable Examples: Making Personal Data Unexploitable. Huang et al. ICLR, 2021.
> [2] Fawkes: Protecting privacy against unauthorized deep learning models. Shan et al. USENIX Security. 2020.
> [3] Poisoning attacks against support vector machines. Biggio et al. ICML, 2012.
> [4] Support vector machines. Hearst et al. IEEE Intelligent Systems and their applications, 1998.
> [5] Towards poisoning of deep learning algorithms with back-gradient optimization. Muñoz-González et al. ACM workshop on artificial intelligence and security, 2017.
> [6] Generative poisoning attack method against neural networks. Yang et al. arXiv preprint arXiv:1703.01340, 2017.
> [7] Targeted backdoor attacks on deep learning systems using data poisoning. Chen et al. arXiv preprint arXiv:1712.05526, 2017.
> [8] Poison frogs! targeted clean-label poisoning attacks on neural networks. Shafahi et al. NeurIPS, 2018.
> [9] Reflection backdoor: A natural backdoor attack on deep neural networks. Liu et al. ECCV, 2020.

---

> > ### Author Response · Authors · 2022-08-01
> > **Answer question 2 (1/1)**
> >
> > **Q2. I suggest, using "perturbed" images instead of "encrypted" images.**
> >
> > **A2.** In this paper, we name the noise-enhanced images as the ''encrypted'' images according to the effect of noises. This nomenclature is common in the field of security machine learning. For example, in adversarial attacks, the "perturbed" images are called ''adversarial'' images (examples), and in traditional data poisoning, the "perturbed" images are called ''poisoned'' images. Therefore, we believe that the term ''encrypted'' images can emphasize that the model trainer cannot exploit useful knowledge from these "perturbed" images. i.e., highlight the unlearnable property of images [1]. We also understand the reviewer's concern. To avoid ambiguity or misleading, we have made two changes in the revised manuscript: 1) Clarify what the term "encryption" means when it first appears in the main text (i.e., Line 37). 2) In the problem statement section (i.e., Line 131), we explicitly illustrate that the ''encrypted'' images in this paper are obtained by adding imperceptible perturbations to the original images.
> >
> > [1] Unlearnable Examples: Making Personal Data Unexploitable. Huang et al. ICLR, 2021.

---

> > > ### Author Response · Authors · 2022-08-01
> > > **Answer question 3 (1/2)**
> > >
> > > **Q3. The current discussion on how attackers can bypass the defence is a bit limited. What happens when some users have both their perturbed and un-perturbed data online? Since ConfounderGAN can be obtained by any user, the attacker can also use it to train a denoiser. Overall, more discussion on the adaptive setting could help.**
> > >
> > > **A3.** We supplement an experiment to explore this problem. To better understand the adaptive setting proposed by [10], we first illustrate the assumption on the data owner’s capability and the model trainer’s capability under this setting:
> > >
> > > **Assumption on data owner’s capability:** A data owner processes some natural images $D_{nat}$ into encrypted images $D_{en}$ via ConfounderGAN and uploads them to social media.
> > > **Assumption on model trainer’s capability:** A model trainer knows that these images $D_{en}$ have been processed by ConfounderGAN and can directly access the generator of ConfounderGAN. The model trainer wishes to train a denoiser against the noise generated by the ConfounderGAN. However, the trainer cannot obtain the original images $D_{nat}$ corresponding to the encrypted images $D_{en}$, otherwise he/she can directly use these original images $D_{nat}$ to train the model. Therefore, in order to denoise the encrypted images $D_{en}$, the model trainer needs to do the following steps: 1) collect additional public images $D_{nat}^\prime$. 2) feed the surrogate images $D_{nat}^\prime$ into the generator of ConfounderGAN to build the encrypted images $D_{en}^\prime$. 3) use $D_{nat}^\prime$ and $D_{en}^\prime$ to train a denoiser. 4) remove the noise of the encrypted images $D_{en}$ by the trained denoiser.
> > >
> > > We conduct the experiment on CIFAR10 to investigate whether the adaptive denoiser can completely remove the effect of the encrypted noises. In practice, we divide the training set of CIFAR10 into two equally as $D_{nat}$ and $D_{nat}^\prime$, and then use the above steps to obtain the denoised images, where the training of the denoiser follows DnCNN [11]. The experimental results are shown in the table below.
> > >
> > > |  Training dataset   | natural images $D_{nat}$  |denoised images  |encrypted images $D_{en}$  |
> > > |  :----:  | :----:  |:----:  |:----:  |
> > > | Test accuracy  | 87.4\% |77.9\% |11.9\% |
> > >
> > > The result shows that although the denoiser can resist our encryption method to a certain extent, the model’s performance can still be significantly compromised by our confounder noises, which shows that our method retains its effectiveness under the adaptive setting.

---

> > > > ### Author Response · Authors · 2022-08-01
> > > > **Answer question 3 (2/2)**
> > > >
> > > > Meanwhile, for the adaptive setting proposed by Radiya-Dixit et al. [10], we think there are two important points that need to be clarified.
> > > >
> > > > **1) We believe that this adaptive setting has unbalanced assumptions about the strength of the data owner’s capability and model trainer’s capability, leaving the data owner (or crypto tool designer) on the weaker side.**
> > > > Radiya-Dixit et al. [10] assume that the model trainer has full access to the encryption tool, while the crypto tool designer has no knowledge of the model trainer's decryption method. For example, in our paper, ConfounderGAN’s designer does not know that the model trainer will use a denoiser to remove encryption noises. However, if we know this information in advance, we might be able to introduce the knowledge of the denoiser into the training process of the ConfounderGAN, making it robust to the denoiser. An intuitive idea is to change the existing training architecture from "original image -> generator -> confounder noise -> pretrain classifier" to "original image -> generator -> confounder noise -> **pretrain denoiser** -> pretrain classifier", so that the confounder property can be preserved even if the generated noises encounter a denoiser in the future. Of course, this solution is very rudimentary. We will explore the optimal solution in future work, thus giving the data owner (or crypto tool designer) an edge in the arms race with the model trainer.
> > > >
> > > > **2) Since data owners usually don't reveal which encryption tool they use, we believe that the non-adaptive setting may be more practical than the adaptive setting in real-world scenarios.**
> > > > Radiya-Dixit et al. [10] believe that the adaptive setting is practical in the real-world, and they give the following argument: encryption tools are usually publicly accessible applications, thus model trainers can adaptively train a feature extractor that resists these encryption noises. We agree that encryption tools are generally publicly accessible, but disagree that model trainers can adaptively train feature extractors. This is because multiple encryption methods will be proposed in the future. When data owners publish encrypted data, they won't reveal the encryption tools they use in most scenarios. Thus it is difficult for the model trainers to determine which encryption method should be used when adaptively training decryption feature extractors. In fact, referring to the community of adversarial examples [12,13], one encryption method can derive multiple instantiations by modifying the encryption constraints. For example, data owners can replace small pixel-wise perturbation with watermark [12], color channel perturbation [13], etc., according to their preferences. As long as the data owner does not expose the information of the encryption tool, the model trainer cannot decrypt it in an adaptive manner. Based on these analyses, we believe that the non-adaptive setting may be more practical than the adaptive setting in real-world scenarios.
> > > >
> > > > We have added experiments and discussions about the adaptive setting in the **Appendix D** of the revised manuscript.
> > > >
> > > > [10] Data Poisoning Won't Save You From Facial Recognition. Radiya-Dixit et al. arXiv preprint arXiv:2106.14851, 2021.
> > > > [11] Beyond a gaussian denoiser: Residual learning of deep cnn for image denoising. Zhang et al. TIP, 2017.
> > > > [12] Adv-watermark: A novel watermark perturbation for adversarial examples. Jia et al. Proceedings of the 28th ACM International Conference on Multimedia. 2020.
> > > > [13] Color channel perturbation attacks for fooling convolutional neural networks and a defense against such attacks. Kantipudi J et al. IEEE Transactions on Artificial Intelligence, 2020.

---

> > > > > ### Author Response · Authors · 2022-08-08
> > > > > **Reminder**
> > > > >
> > > > > Dear Reviewer. Have you had a chance to look at our rebuttal and updated paper? We're eagerly awaiting your response to better address your concerns.

---

> > > > > ### Comment · Reviewer_oeZM · 2022-08-08
> > > > > **Update**
> > > > >
> > > > > Thank you for the update.
> > > > >
> > > > > Based on the additional experiments performed, the paper update, and the discussion, I am willing to update my score to weak accept.
> > > > >
> > > > > I am still unsatisfied with the discussion on the adaptive settings.
> > > > >
> > > > > First, no matter how well ConfounderGAN is designed it should be assumed that the adversary can have access to it as well. Second, the adaptive attack is posterior to the release of ConfounderGAN. Assuming that data owners won't reveal the encryption tools they use in most scenarios is a form of security through obscurity, which violates the Kerckhoffs's principle that any good "encryption" system should follow.

---

> > > > > > ### Author Response · Authors · 2022-08-09
> > > > > > **Thanks for your suggestion**
> > > > > >
> > > > > > Thanks for your suggestion, although we believe that the current version of ConfounderGAN can be applied to most practical scenarios (i.e. as data owners usually don't reveal which encryption tool they use), we also note that it is important to design an encryption tool that strictly satisfies the Kerckhoffs's principle [14]. In future work, we will further improve ConfounderGAN so that it can work optimally under this principle.
> > > > > >
> > > > > > Meanwhile, we also add the above discussion to Appendix E (Line 573 – Line 576).
> > > > > >
> > > > > > [14] La cryptographie militaire. Petitcolas F. J. des Sci. Militaires, 1883.

---

### Author Response · Authors · 2022-08-01
**Summary of Changes in Revision**

We thank all reviewers for the detailed comments and constructive suggestions, which have undoubtedly improved the quality of our manuscript. We have uploaded the revised manuscript and appendix based on the reviewers’ feedback, and have highlighted changes from the original submission in red. We summarize the notable changes below, and refer to minor changes in the individual responses.

* As requested by Reviewer oeZM and rw45, we have discussed the relation between our method and **data poisoning in related work**.
* As requested by Reviewer oeZM and Bo5j, we have conducted the experiment to investigate the effectiveness of our method under the adaptive setting and given a detailed discussion about this setting in **Appendix D**.
* As requested by Reviewer Bo5j, we add the performance comparison of our method and DeepConfuse under the out-of-distribution encryption setting in **Appendix E**.

---

### Meta-Review · Area_Chair_R88v · 2022-09-06

**Recommendation:** Accept
**Confidence:** Certain

**Metareview:**

All the reviewers were excited by the idea and a efficient method to solve very critical problem with rigorous experimental support. They all agreed that the paper is above bar for publications. We hope the authors will further improve the paper for camera ready submission.

**Award:**

No

---

### Decision · Program_Chairs · 2022-09-14

Accept